# Fault-Aware Neural Code Rankers

**Jeevana Priya Inala**        **Chenglong Wang**        **Mei Yang**        **Andres Codas**

**Mark Encarnación**        **Shuvendu K Lahiri**        **Madanlal Musuvathi**        **Jianfeng Gao**

Microsoft Research
{jinala,chenwang,meiyang,andres.codas,markenc,
shuvendu,madanm,jfgao}@microsoft.com

## Abstract

Large language models (LLMs) have demonstrated an impressive ability to generate code for various programming tasks. In many instances, LLMs can generate a correct program for a task when given numerous trials. Consequently, a recent trend is to do large scale sampling of programs using a model and then filtering/ranking the programs based on the program execution on a small number of known unit tests to select one candidate solution. However, these approaches assume that the unit tests are given and assume the ability to safely execute the generated programs (which can do arbitrary dangerous operations such as file manipulations). Both of the above assumptions are impractical in real-world software development. In this paper, we propose CODERANKER, a neural ranker that can predict the correctness of a sampled program without executing it. Our CODERANKER is fault-aware i.e., it is trained to predict different kinds of execution information such as predicting the exact compile/runtime error type (e.g., an IndexError or a TypeError). We show that CODERANKER can significantly increase the pass@1 accuracy of various code generation models (including Codex [11], GPT-Neo, GPT-J) on APPS [25], HumanEval [11] and MBPP [3] datasets.

## 1 Introduction

Large transformer-based language models (LLMs) have impressive capabilities [19, 7, 16], including the ability to generate code [11, 3, 28, 39, 33]. The task here is to take a natural language description or previous code context as the input and generate an entire program in a general-purpose programming language such as Python or C++. Generating entire programs is a challenging task as it involves understanding the task, figuring out how to accomplish it, and writing code without any syntax/runtime errors. On harder programming problems such as coding competition problems, current models achieve very low accuracy especially if the inference-time sampling budget is low. For example, on the APPS dataset [25], a state-of-the-art code generation model, Codex [11], achieves 4% accuracy if it is allowed to sample only one program per task (called pass@1), but achieves 24% accuracy if it is allowed 100 samples per task (called pass@100—at least one correct program in 100 samples) (see Table 7). This observation leads us to an important research problem on exploring approaches to rank the sampled programs to bridge the gap between the pass@1 and pass@100 performances.

Upon analyzing the sampled programs obtained using an LLM, we find that several of them have syntax errors and runtime errors, and some of them execute to produce undesirable outputs. Therefore, prior works [11, 28] focused on ranking the programs by executing them on a small set of unit tests (which are typically assumed to be given as part of the task description). However, there are several caveats to this approach: First, even if a program passes the given tests, it could still fail on the unknown unit tests. Second, there is a burden on the user to provide unit tests for every inference

36th Conference on Neural Information Processing Systems (NeurIPS 2022).

| Task | Reference code | Unit tests |
|------|----------------|------------|
| Write a function name 'nextPerfectSquare' that returns the first perfect square that is greater than its integer argument. Example: next_perfect_square(6) == 9 def next_perfect_square(n): Use Call−Based format | ```python def next_perfect_square(n): return n>=0 and (int(n**0.5)+1)**2 ``` | Inputs: 6 36 0 −5 Outputs: 9 49 1 0 |

**Figure 1:** An example code generation task from the APPS dataset (train/3447).

task. Third, to execute the code, all the dependencies has to be installed properly. This is especially problematic in scenarios where a user wants to get code suggestions in a project with multiple files and dependencies (such as using CoPilot in a VS Code environment [1]). Even if these dependencies are satisfied, many realistic programming scenarios involve incomplete code under active development where execution is just infeasible. Finally, the code generated by a LLM could potentially have security risks (such as deleting files on the disk) and hence, executing such a code needs heavy-weight isolation mechanisms such as a sandbox environment.

To alleviate the above issues with relying on executing code, some of the recent works [18, 36] proposed to use a neural-network based ranker[1] for ranking programs sampled from a LLM. A ranker model is essentially a classifier that takes as input a task description and a sampled program and predicts the probability that the program is correct with respect to the task description. When given multiple programs (obtained by sampling), a ranker re-orders them in the decreasing order of the predicted probability of the program being correct. The ranker is essentially trying to emulate executing a program on some unit tests without actually executing the code during inference. The training data is obtained by executing both correct and wrong programs sampled from the code generation model itself. Thus, we only need unit tests and the execution ability during the dataset creation step instead of during inference as in prior approaches.

While previous approaches target math problems to make the learning problem tractable, in this paper, we design CODERANKER for more complex general programming tasks in Python. A sampled Python program can fail in many different ways. For example, a program when executed on a unit test can result in a compile/runtime error and can produce a wide variety of outputs such as integers, strings, lists and dictionaries that can produce a type mismatch. In contrast in the math domain, there is less scope for compile/runtime errors and the output is usually just a number.

Our CODERANKER approach is based on the idea that *a neural ranker trained to distinguish between the various failure modes can have a better understanding of the program and the task, and hence can do better at ranking programs.* Thus, we design *fault-aware* CODERANKER and we investigate its impact on the ranking performance. Each fault-aware ranker is a classifier trained to predict one/two multi-class labels that are extracted from the rich information obtained by executing the programs.

We used the APPS dataset to finetune/train the code generation and the CODERANKER models. On this dataset, we showed that CODERANKER improved the pass@1 performance of Codex (used in a few-shot manner) from 26% to 39.6% on the validation set and from 3.8% to 4.5% on the test set. We additionally found our rankers are transferable to a different dataset without any additional training. On the HumanEval dataset, we improved Codex's pass@1 from 26% to 32% and on the MBPP dataset, we improved from 36% to 42%. We found similar performance boosts with other code generation models such as GPT-J and GPT-Neo. Compared with a naïve binary classifier-based ranker, our fault aware CODERANKER achieves better ranking performance. Finally, we investigated the effect of mixing ranker datasets from multiple code generation models which gives us an extra boost in the performance.

## 2 Preliminaries

### 2.1 Code generation

**Task:** A code generation task $G$ is a prompt, represented as a sequence of tokens, that specifies the task at hand. $G$ is usually a combination of natural language, input-output examples, and starter

---

[1]called a verifier in the previous work

| Model | # parameters | mode | pretrain dataset | finetune dataset |
|---|---|---|---|---|
| Codex (Davinci) | Unknown | One-shot | GitHub + Common Crawl, Wiki, etc. | N/A |
| GPT-J | 6B | Fine-tuned | Pile dataset | APPS train dataset |
| GPT-neo 1.3B | 1.3B | Fine-tuned | Pile dataset | APPS train dataset |
| GPT-neo 125M | 125M | Fine-tuned | Pile dataset | APPS train dataset |

**Table 1:** Code-generation models, their sizes, the mode of usage (fine-tuned or few-shot), and the datasets used for pretraining and finetuning.

code. A solution $S$ to a code generation task is a sequence of tokens that together form a program to solve the given task. Existing datasets additionally contain a set of input-output pairs that are used to test the correctness of a generated program. Figure 1 gives an example code generation task, the corresponding reference solution, and the unit tests taken from the APPS dataset [25].

**Models:** There are several existing pre-trained LLMs in the literature that are suitable to generating code either in a few-shot manner or after finetuning. A code generation model $F$ provides a way for us to sample programs for a given task $G$ as $S_i \sim P_F(S \mid G)$ where $P_F$ denotes the probability distribution induced by the code generation model $F$.

In this paper, we study four different code generation models—(i) Codex, (ii) GPT-J (6B), (iii) GPT-Neo 1.3B and (iv) GPT-Neo 125M (Table 1). Codex is the largest state-of-the-art code generation model that is publicly available for querying through an API and has shown impressive performance in code generation [11]. It is built on top of a GPT-3 language model architecture and is trained on 180 GB of GitHub data. GPT-Neo and GPT-J are open-sourced models with the number of parameters ranging from 125M to 6B. These models are pre-trained on the Pile dataset (800 GB of natural language corpus with 8% of GitHub data). Since, these models are open-sourced, it is possible to finetune these models on a downstream dataset such as the APPS dataset. In addition to the above models, there are other code-generation models such as AlphaCode [28] and Google's model [3]. While our approach is applicable to any of these models, in this study, we validate the effectiveness of CODERANKER on a set of state-of-the-art code generation models that are publicly available, for reproducibility.

**Metrics:** Code generation models are evaluated based on functional correctness rather than exact/-fuzzy match to a reference program. This is because match-based metrics are unable to account for the large and complex space of programs functionally equivalent to a reference program. Functional correctness is estimated by checking whether a sampled program passes a set of unit tests. Prior approaches evaluate functional correctness using the pass@k metric; given $k$ generated program samples per task, a task is considered solved if any of the samples passes the unit tests. The pass@k metric measures the total fraction of tasks solved. Additionally, we define the exec@k metric which computes the fraction of the tasks for which there exists at least one program in the $k$ samples that executes without any compile/runtime errors, i.e., produces a non error value for each of the test inputs, but may or may not match the desired output.

## 2.2 Code ranking

Given $n$ sampled programs using a code generation model, $S_1, \cdots, S_n \sim P_F(S \mid G)$, the goal of CODERANKER is to find an ordering of the programs $S_{o_1}, \cdots, S_{o_n}$ such that to compute the *ranked pass@k* for $k \leq n$, a problem is considered solved if any program in the set $\{S_{o_1}, \cdots, S_{o_k}\}$ passes the unit tests. A CODERANKER model $R$ takes as input a code generation task $G$ and a sampled program $S_i$ and outputs a score $s_i$. The ranked ordering of the sampled programs is given by $S_{o_1}, \cdots, S_{o_n}$ such that $s_{o_1} > s_{o_2} > \cdots > s_{o_n}$.

## 3 Fault-Aware Neural Code Ranker

A CODERANKER model is a classifier that is trained to classify a pair $\langle G, S_i \rangle$ as CORRECT or not, where CORRECT means that $S_i$ satisfies the task $G$ with respect to its unit tests. The score $s_i$ is computed as $s_i = P_R(\text{CORRECT}|G, S_i)$ where $P_R$ is the probability according to the ranker model $R$. This probability is extracted using the real values from the last layer before the SoftMax layer.

| Code generation model used for generating the dataset | Training data | | | Validation data | | | Test data | | |
|---|---|---|---|---|---|---|---|---|---|
| | C | W | | C | W | | C | W | |
| | | I | E | | I | E | | I | E |
| Codex | 70K | 180K | 140K | 16K | 26K | 18K | 19K | 280K | 200K |
| GPT-J | 20K | 210K | 150K | 3K | 32K | 23K | 2.4K | 210K | 260K |
| GPT-Neo 1.3B | 10K | 170K | 200K | 1.5K | 28K | 27K | 0.7K | 170K | 310K |
| GPT-Neo 125M | 10K | 150K | 220K | 0.8K | 23K | 34K | 0.2K | 140K | 340K |

**Table 2:** Ranker dataset distribution for the datasets generated using the 4 different code generation models on the APPS tasks. C: # CORRECT programs, W: # WRONG programs, I: # intent errors, and E: # execution errors.

| Generated Program | Full error message | Labels |
|---|---|---|
| ```python
def number_property(n):
    if n % 2 == 1: return True
    return  random.choice(n) % 1 == 0
``` | *TypeError("object of type 'int' has no len()") at Line 2* | B: wrong
T: execution  error
I: —
E: TypeError
L: Line 2 |
| ```python
def gematria( string ):
    return  ''. join (sorted ( string . lower (),
      key=lambda item: item.lower(),
      reverse=True)).lower()
``` | *Expected output is 775, but generated output is 'vole'* | B: wrong
T: intent  error
I: OutputTypeError
E: —
L: −1 |

**Table 3:** Examples from the fault-aware ranker dataset showcasing the generated program, the full error message obtained by executing the program, and the fine-grained labels extracted from this error message. Each entry contains a binary label (B) that classifies the entry as CORRECT or WRONG, a ternary label (T) that distinguishes between CORRECT, intent error, and execution error, an intent error label (I) that specifies the type of the intent error (or -), an execution error label (E) that specifies the type of the execution error (or -), and a line number (L) that corresponds to the erroneous line in the code (or -1 if there is no execution error).

## 3.1   Code Ranker Dataset

To train a CODERANKER model, we need a dataset that has both CORRECT and WRONG (i.e., not correct) programs. To collect these programs, we use the code-generation models to sample $n = 100$ programs for each task in the training dataset. We then execute the sampled programs on their corresponding unit tests to generate the classification labels. Following the observations from [18], one has to be careful to not over-train the base code generation models to ensure diversity in the sampled programs. Hence, we only finetune the base code generation models for a maximum of 2 epochs and chose a checkpoint that results in the lowest validation loss (this does not apply to the Codex model, which is used in a few-shot manner). Table 2 shows the distribution of the ranker's datasets obtained by using the 4 different code generation models for the tasks in the APPS dataset [2]. As one can expect, the ranker dataset is highly imbalanced with about 5X to 40X more WRONG data points than CORRECT data points and this ratio is higher for smaller models such GPT-Neo models.

**Fault-aware ranker dataset:** A straightforward ranker is one that is trained to predict a binary label CORRECT or WRONG. However, a program fails for various reasons and knowing why a program might fail is crucial to predicting whether a program is CORRECT or not. Therefore, we designed a fault-aware ranker dataset. When we execute a program on a set of unit tests, the compiler message is more than just a single bit of information. In fact, when the unit test fails, we know if it failed because of a compile/runtime error (which we call an *execution error*) or because the program produced a wrong output for a particular input (which we call an *intent error*). Table 2 also shows the distribution of intent errors and execution errors in the ranker datasets. An interesting observation is that the percentage of execution errors decreases for larger code generation models.

It is possible to further break down the WRONG datapoints. Within the execution error class, the compiler message tells us exactly the type of the execution error (such as an IndexError or a TypeError or a TimeOutError) and the line in the program that caused this error. By parsing the error message from the Python compiler, we derived 10 most frequent classes of execution errors as shown in Table 4 and Figure 2. Similarly, for the intent error class, we know how the generated output differs from the expected output (such as wrong type or wrong length of the array output). We manually

---

[2]We used a subset of 600 problems as the validation set, chosen randomly from a subset of the original training problems for which the best model, Codex, produces at least one correct program in its 100 samples.

| Class | Description |
|---|---|
| NameError | Undefined variables |
| ValueError | Operation/function received an argument of inappropriate value |
| EOFError | Raised because of extra input() functions |
| TypeError | An operation or function is applied to an object of inappropriate type |
| IndexError | Array index out of bounds |
| KeyError | Key in a dictionary is not found |
| TimeoutException | Code is inefficient or doesn't terminate |
| SyntaxError | Parser encountered an error |
| Function not found | The function expected by the unit tests is not found |
| Misc | Other execution errors |

**Table 4:** Different classes of execution errors. See `https://docs.python.org/3/library/exceptions.html` for more details.

(a) NameError

```
def bird_code(arr):
   if 'hyphen' in arr:
     return [arr[i][0] + ...
# name 'i' is not defined in line 2
```

(b) KeyError

```
def league_standings(teams):
   return {i+1: teams[−i−1] for
     i in range(len(teams))}
# KeyError(−1) at Line 1
```

(c) TimeoutException

```
def fibonacci(n):
   return n if n in [0, 1] else
     fibonacci(n − 1) +
     fibonacci(n − 2)
# inefficient implementation
```

**Figure 2:** Examples of execution errors.

designed 9 most frequent classes of intent errors by looking at various expected and generated outputs from the training examples (see Table 5 and Figure 3). These fine-grained execution-based labels constitute our fault-aware dataset. Table 3 shows a few data entries with all the labels. Figure 6 in the Appendix shows the distribution of the various classes of execution errors and intent errors for the ranker dataset obtained using the Codex model.

## 3.2 Code Ranker Tasks

We now describe the different classification tasks derived from the above dataset. For each classification task, the input is a pair of code generation task specification $G$ and a generated program $S_i$ and the output is one or multiple labels where each label belongs to pre-determined set of classes. We describe the different labels/classes of the various tasks below. These tasks are designed to explore the trade-offs between having abstract classes of failures versus having fine-grained classes of failures.
**Binary (B):** The output is a single binary label with two classes {CORRECT, WRONG}.
**Ternary (T):** The ternary task splits the WRONG class into intent error and execution error classes: this forms a three-class classification task with output labels {CORRECT, intent error, execution error}
**Intent Error aware (I):** The intent error aware task splits the intent error class in the ternary task into its 9 different sub-classes, thus has a total of 11 classes for the output label.
**Execution Error aware (E):** The execution error aware task is similar to the intent error aware task but instead of splitting the intent error class, we now split the execution error class into its 10 different sub-classes, thus has a total of 12 classes for the output label.
**Execution Error + Error Line aware (E+L):** This is a multi-class and multi-label classification task which combines two classification tasks. The first one is the execution error aware task described above. The second task is to predict the line of the code that corresponds to an execution error. The labels for the error line number also includes −1 to represent there is no execution error.

## 3.3 Code Ranker Models

We implemented our fault aware CODERANKER models by finetuning the pretrained CodeBERT [21] model.

**CodeBERT:** It is a state-of-the-art pretrained BERT-style code-understanding model trained on the CodeSearchNet dataset [26] using a combination of masked language modeling and replaced token detection objectives [17]. It takes as input a concatenation of two segments with a special separator token, namely $[CLS], w_1, w_2, ..w_n, [SEP], c_1, c_2, ..., c_m, [EOS]$. Usually, one segment is a natural language text, and another is a code. [CLS] is a special token, whose final hidden representation can be treated as the aggregated sequence representation for classification or ranking downstream tasks.

**Adding a classification head:** We add a classification head on top of a base CodeBERT model by connecting a linear layer and a softmax layer to the hidden representation of the [CLS] special token. Let $C \in \mathbb{R}^H$ be the final hidden vector corresponding to the [CLS] token and let $W \in \mathbb{R}^{K \times H}$ be the

| Class | Description |
|---|---|
| NoneError | Generated output is None while expected is not |
| EmptyError | Generated output is an empty array while expected is not |
| OutputTypeError | Different types of outputs |
| LengthError | Arrays/Dicts/Sets of different lengths |
| IntSmallError | Integer outputs that are different from the expected ones with delta $\leq 10$ |
| IntLargeError | Integer outputs that are different from the expected ones with delta $> 10$ |
| StringSmallError | String outputs whose length is different from the expected ones with delta $\leq 3$ |
| StringLargeError | String outputs whose length is different from the expected ones with delta $> 3$ |
| Misc | Other intent error |

**Table 5:** Intent error classes.

(a) NoneError

```
def consecutive_sum(num):
    sum = 0
    n = len(str(num))
    for i in range(n−1):
        if num % i == 0 and sum > 0:
            return sum
# return statement never gets triggered
```

(b) LengthError

```
def diamonds_and_toads(sentence,fairy):
    return dict(zip('Ruby␣Crystal',
            (0, 2, 1, 2, 0)))
# Got {'R': 0, 'u': 2, 'b': 1, 'y': 2},
# expected {'ruby': 3, 'crystal': 2}
```

(c) StringSmallError

```
def smash(words):
    return ''.join(word for word in words)
# Got 'helloworld',
# expected 'hello world'
```

**Figure 3:** Examples of intent errors.

| Code gen. model | pass@100 | pass@1 | ranked pass@1 | pass@5 | ranked pass@5 | exec@1 | ranked exec@1 |
|---|---|---|---|---|---|---|---|
| Codex | 100 | 26 | 39.6 | 56.4 | 63.5 | 69.7 | 87.0 |
| GPT-J | 48.8 | 5.1 | 11.0 | 15.6 | 21.7 | 60.4 | 82.9 |
| GPT-Neo 1.3B | 34.6 | 2.6 | 8.0 | 9.1 | 15.1 | 52.1 | 85.6 |
| GPT-Neo 125M | 23.6 | 1.4 | 6.5 | 5.2 | 11.4 | 41.1 | 58.9 |

**Table 6:** Results on the APPS validation dataset about how our fault-aware rankers can improve the pass@1, pass@5, and exec@1 performance for various code generation models. We use the best ranker model for each code generation model for this result. Pass@100 for the Codex model on this validation set is 100% by design.

weights of the newly added classification layer where $K$ is the number of classes in the classification task. The logits for the classification output are computed as softmax($CW^\top$) and we use a standard cross-entropy loss for finetuning all the weights.

**Adding a line prediction head:** To predict the line corresponding to an execution error, we introduce an error line vector $S \in \mathbb{R}^H$ during fine-tuning. The probability of a newline ("\n") token $i$ being the erroneous line is computed as a dot product between $T_i$ and $S$ followed by a softmax over all of the newline tokens in the code, i.e., $P_i = \frac{e^{ST_i}}{\sum_j e^{ST_j}}$ where $T_i$ is the final hidden vector corresponding to the $i$th newline token. We include a newline token at the beginning and the end of the input to indicate the case where there is no erroneous line in the code (i.e., code does not result in an execution error) and to indicate the case where the erroneous line is beyond what is encoded in the input (this occurs if the task+code context cannot fit within the 512 token limit of CodeBERT), respectively.

## 4 Evaluation

We next evaluate our CODERANKER approach. We investigate (1) how fault-aware CODERANKERs can improve various code generation models on various code datasets, (2) the impact of the different

| Code gen. model | pass@100 | pass@1 | ranked pass@1 | pass@5 | ranked pass@5 | exec@1 | ranked exec@1 |
|---|---|---|---|---|---|---|---|
| Codex | 24.1 | 3.8 | 4.5 | 9.2 | 10.2 | 59.6 | 73.4 |
| GPT-J | 7.2 | 0.5 | 0.8 | 1.6 | 2.6 | 45.4 | 63.8 |
| GPT-Neo 1.3B | 3.0 | 0.14 | 0.3 | 0.53 | 1.1 | 35.2 | 73.7 |
| GPT-Neo 125M | 1.5 | 0.04 | 0.1 | 0.17 | 0.5 | 28.5 | 43.9 |

**Table 7:** Results on the APPS test dataset. We use the best checkpoints for the code generation models and the ranker models based on the results on the validation set.

ranker tasks, and (3) the effect of mixing ranker datasets generated by different code generation models.

## 4.1 Experiment Setup

**Code generation datasets:** We consider three existing code generation datasets for our evaluation: (1) APPS [25]: a collection of 5000 training and 5000 test tasks collected from coding competitions and interview problems, (2) HumanEval [11]: a set of 164 test tasks, and (3) MBPP [3]: a set of 974 mostly basic python programming tasks with 474 training problems and 500 test problems.

In our experiments, we only use the APPS dataset for finetuning the code generation models and the CODERANKER models (since it is the largest dataset). But we evaluate these models on all three sets of test tasks. The APPS dataset does not come with a validation dataset, so we used a set of 600 tasks from the original training dataset for validation; these are, then, excluded from the training dataset. Since we are interested in explicitly evaluating the ability of a ranker to distinguish CORRECT code from WRONG code, we chose our validation set to only include problems for which a Codex model [3] (in a few-shot manner) can generate at least one correct program in its 100 samples. To facilitate the transfer of GPT-J and GPT-Neo finetuned models on the HumanEval and the MBPP datasets, we perform a minor programmatic transformation of the task descriptions to match the APPS style.

**Metrics:** We use the pass@1, pass@5, exec@1, ranked pass@1, ranked pass@5, and ranked exec@1 metrics (higher values are better). See Section 2 for their definitions. We also show the pass@100 metric to illustrate the maximum possible value for the pass@k and ranked pass@k metrics. These metrics are measured using an unbiased estimator from 100 samples as proposed by [11].

**Training setup and hyper-parameters:** We finetuned GPT-J and GPT-Neo code generation models on the APPS training dataset for 2 epochs with a batch size of 256 and a learning rate of 1e-5, and chose the checkpoint that has the lowest validation loss. For inference, we used temperature sampling with $T = 0.8$ for Codex model and $T = 0.9$ for the GPT-J and GPT-Neo models unless specified otherwise. We chose these temperatures to maximize diversity in the 100 samples, but we also conduct an ablation with lower temperatures in Table 11. For each program, we sample 512 new tokens and truncate the generated program by a special stop sequence that we used in the few-shot/finetuning prompts.

We finetuned the CODERANKER models for 30 epochs with a batch size of 512 and a learning rate of 1e-4, and chose the checkpoint that results in the best ranked pass@1 metric on the validation dataset. We used class weights to balance the different classes while training the rankers. All experiments are conducted on V100-32GB GPUs.

**Notation**: We use the notation $R_{D_X}^Y$ to denote a CODERANKER model trained on a dataset obtained using the code generation model $X$ from Table 2 and on one of the five ranker tasks $Y$ from Section 3.2.

## 4.2 Main Results: CODERANKER improves code generation models

**APPS validation dataset:** First, we analyze the results on the APPS validation dataset of 600 tasks. Table 6 shows the performance on various metrics for the 4 different code generation models. These results use the best CODERANKER model for each code generation model which is shown in Table 10. From Table 6, we find that CODERANKERs improve all the metrics for all the code generation models despite their different sizes. The pass@1 performance increases by 5.1% to 13.6% with CODERANKER and the models can solve about 30 to 80 more tasks when it has to select only one program from the 100 samples. Another interesting observation is that a GPT Neo 125M model when combined with CODERANKER (another 125M model) beats a GPT-J model (with 50X more parameters). These results show the effectiveness of CODERANKER on improving the code generation models.

**APPS test dataset:** Our results on the APPS test dataset of 5000 tasks is shown in Table 7. The test problems are harder than the ones on the validation set, which we can see by the smaller pass@100

---

[3]We chose Codex for this filtering step because it is the best performing model out of the 4 models we considered.

| Code gen. model | pass@100 | pass@1 | ranked pass@1 | pass@5 | ranked pass@5 | exec@1 | ranked exec@1 |
|---|---|---|---|---|---|---|---|
| Codex | 88.4 | 26.3 | 32.3 | 50.5 | 61.6 | 77.0 | 86.6 |
| GPT-J | 45.1 | 9.1 | 11.6 | 19.1 | 18.9 | 73.6 | 89.0 |
| GPT-Neo 1.3B | 19.5 | 3.2 | 6.1 | 7.7 | 8.5 | 66.3 | 87.8 |
| GPT-Neo 125M | 12.8 | 0.84 | 3.0 | 3.0 | 6.1 | 52.3 | 58.3 |

**Table 8:** Results on the HumanEval dataset showing the zero-shot transferability of the rankers trained on APPS.

| Code gen. model | pass@100 | pass@1 | ranked pass@1 | pass@5 | ranked pass@5 | exec@1 | ranked exec@1 |
|---|---|---|---|---|---|---|---|
| Codex | 84.8 | 36.4 | 41.8 | 60.9 | 62.4 | 74.8 | 93.2 |
| GPT-J | 60.4 | 12.1 | 14.2 | 28.9 | 28.2 | 73.3 | 80.8 |
| GPT-Neo 1.3B | 40.0 | 3.6 | 5.0 | 11.9 | 16.2 | 58.3 | 75.2 |
| GPT-Neo 125M | 6.8 | 0.2 | 0.8 | 0.9 | 2.6 | 7.6 | 14.2 |

**Table 9:** Results on the MBPP dataset, showcasing another instance where our fault-aware rankers can be used in a zero-shot manner on a different dataset.

and pass@1 numbers. Hence, the improvement from CODERANKER is smaller in scale, but it is still a significant improvement; Codex's pass@1 increases from 3.8% to 4.7% (35 additional problems).

**HumanEval and MBPP datasets:**  We measure the transfer ability of CODERANKER on two different datasets–HumanEval (results in Table 8) and MBPP (results in Table 9). We can again see that CODERANKERs improve the performance of all code generation models on all metrics for both datasets (one exception is pass@5 for GPT-J on MBPP). Codex model's pass@1 performance increases by 6% on both datasets. These results show the ability of CODERANKER to transfer out-of-distribution and also showcases that the errors made by the code generation models on different tasks are universal.

**Exec@1 vs pass@1:**  In all the above results, the improvements in the exec@1 metric are higher than the improvements in the pass@1 metric; this shows that CODERANKER is better at identifying execution errors in code than intent errors, which is expected since executing code is shown to be an inherently hard task for language models [3, 34].

### 4.3   Ablations

**Effect of sampling temperature:**  On the HumanEval and the MBPP datasets, we additionally experiment with different temperatures for sampling the 100 programs (see Table 11). As expected, we noticed that pass@100 decreases with lower temperatures, but pass@1 increases. We found that CODERANKER further increases the pass@1 performance for 3 out of the 4 setups and achieved 42.7% ranked pass@1 on the HumanEval dataset—the best known result so far on this dataset [4] [16]. On the MBPP dataset, under a low-temperature sampling setup, we found that CODERANKER slightly decreases the pass@1 performance—this we attribute to the small difference between the pass@1 and the pass@100 metrics, which makes it hard for a learned ranker to beat a random ranking scheme.

**Analyzing different ranker tasks:** Figure 4 shows 4 training curves; one for each code generation model, $X$, showcasing the validation ranked pass@1 curves for the model $X$ when combined with the 5 different rankers $R_{D_X}^*$. Results in a tabular format can be found in the Appendix. From the curves, we can notice that for larger models such as Codex and GPT-J, the rankers trained on the ternary classification task $R^T$ perform the best, and for smaller models such as GPT-Neo 1.3B and GPT-Neo 125M, the ranker trained on the intent-aware classification task $R^I$ and the ranker trained on the execution-aware + error line classification task $R^{E+L}$ perform the best, respectively. We can also notice that the binary rankers $R^B$ perform significantly worse especially with smaller models. These results suggest that when training rankers on a dataset that has more WRONG code, harder classification tasks act as better regularizes. Figure 7 in the Appendix shows similar training curves for the ranked exec@1 metric; here, we can see that $R^E$ and $R^{E+L}$ always achieves best ranked exec@1 because they are trained to identify execution errors and $R^B$ and $R^I$ have the worst performance.

---

[4]this excludes approaches that use execution during inference

| Code gen. Model | Best Ranker |
|---|---|
| Codex | $R^T_{D_{\text{Codex}}}$ |
| GPT-J | $R^T_{D_{\text{GPT-J}}}$ |
| GPT-Neo 1.3B | $R^{E+L}_{D_{\text{GPT Neo 1.3B}}}$ |
| GPT-Neo 125M | $R^I_{D_{\text{GPT Neo 125M}}}$ |

**Table 10:** Best CODERANKER model for each code generation model based on best ranked pass@1 on the validation dataset.

| Setup | pass@100 | pass@1 | ranked pass@1 |
|---|---|---|---|
| HumanEval, Temp=0.8 | 88.4 | 26.3 | 32.3 |
| HumanEval, Temp=0.2 | 69.5 | 35.2 | **42.7** |
| MBPP, Temp=0.8 | 84.8 | 36.4 | 41.8 |
| MBPP, Temp=0.2 | 70.0 | **47.2** | 46.8 |

**Table 11:** Results with different sampling temperatures for the Codex model with rankers on HumanEval and MBPP.

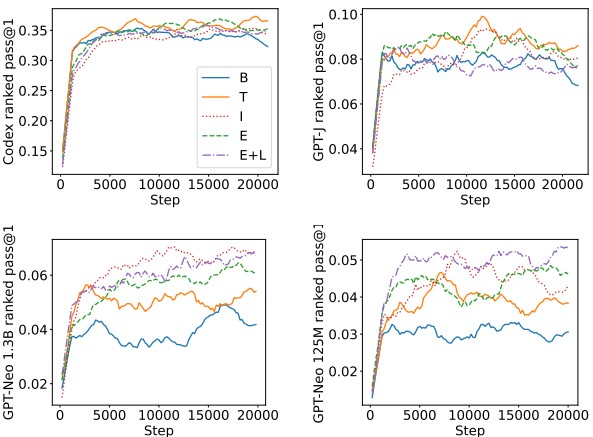

**Figure 4:** Training curves (smoothed) measuring ranked pass@1 over training steps for various rankers on different code generation models.

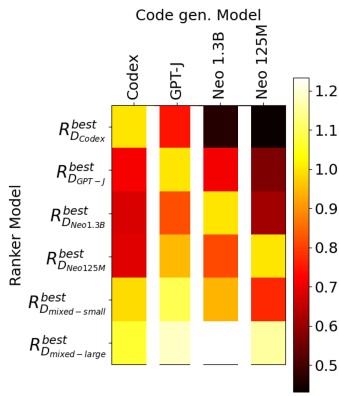

**Figure 5:** Results showing how the ranked pass@1 changes with different ranker datasets.

**Analyzing different ranker datasets:** In this experiment, we measure the impact of using the rankers trained on data from one code generation model on another code generation model and the effect of mixing these ranker datasets. We have 4 different code generation models in this experiment and hence, 4 different ranker datasets. We analyze two different mixed datasets—(i) mixed-small that randomly samples 25% of the above ranker datasets and combines them, and (ii) mixed-large that combines all of the 4 ranker datasets into one; the former represents a dataset that is roughly the same size of the individual ranker datasets for a fair comparison, while the latter makes use of all the available data. Figure 5 shows the ranked pass@1 of all the 4x6 combinations in the form of a heat map. Each column has been normalized such that the values for the same model is 1 (i.e. the diagonal values are 1). From the figure, among the individual ranker datasets, we notice that the rankers trained with the data from the same code generation model are the best and that the rankers trained with data from code generation models with a large size difference are the worst. Finally, the rankers trained on mixed-small dataset perform slightly worse (except for GPT-J code gen. model) than using the same model dataset, but the rankers trained on mixed-large dataset have the overall best performance. These results suggest that while it is usually better to use the same model for generating the ranking dataset, we can potentially use smaller models (less expensive models) to augment the ranker dataset to further improve the performance.

Additional qualitative/quantitative analysis of the CODERANKER approach can be found in the Appendix.

# 5   Related works

**Code generation tasks/models:** There are several code generation models explored in the literature, ranging from decoder-only architectures [11, 3, 39, 6, 22] to encoder-decoder architectures [28, 33, 2, 38] of various sizes. Similarly, there are several task datasets from multiple domains, including math-word-problems [3, 18], Jupyter notebook cell generation [10], common programming tasks [3, 11, 35] and competition level programming problems [28, 25]. In this paper, we evaluate CODERANKER's ability to improve the performance of four decoder-based code generation models

on three programming datasets. Our approach is agnostic to code generation model architectures as long as they generate outputs by sampling and our approach can be easily extended to different domains.

**Code understanding tasks/models:** Besides code generation tasks, there are several code understanding tasks including code search [4, 8, 23, 26], clone detection [37, 30], code summarization [26], code translation [13, 27, 32] and defect detection [9, 5, 15, 40, 5]. Encoder-only code understanding models [29, 21, 2, 38, 24] are developed for these tasks. Our CODERANKER task can be viewed as a new code understanding task, and our ranker model is finetuned from CodeBERT [21]. The defect detection datasets are closely related to ours; the main difference is that prior work focused on finding vulnerability in human-written code, while we focus on detecting errors in model-generated code.

**Filtering/ranking for code generation models:** Previous works such as [11, 28] use execution to prune code completions during inference. While [11] only uses unit tests provided as part of the task, [28] additionally uses a neural model to generate inputs for the unit tests and uses execution to filter out programs that produce the same outputs on those inputs. However, these work requires executing potentially vulnerable code for every inference task. Our approach bypasses execution at inference time with the neural ranker. Similar to our work, [18, 36] propose a neural-network based ranker/verifier for code generation models; the main difference is the domain (general purpose programming tasks in our case versus math problems in prior work) and we train our neural rankers to learn why/how a code fails rather than just predicting a binary label. Moreover, [18, 36] use generative models as their ranker base; in our work, we show benefits of rankers using a simple encoder-only model with only 125M parameters ( [18] uses models with atleast 3B parameters) (see Section A.2.3 in the Appendix to see our ablation on different ranker architectures).

**Using execution/static analysis in other contexts:** There has been other works on using execution to guide code generation by conditioning the generation on a representation of the program states [12, 20]. There are also attempts to replace the execution process with a neural model in these cases [14, 34]. Another relevant work is [31], which improves the code generation models by using static analysis to augment the model's input and has shown to drastically reduce the number of execution errors made by the generated programs. This paper takes an alternate approach by using a neural ranker model to prune out wrong programs at inference time.

# 6    Conclusions and Future Directions

We presented CODERANKER that ranks programs generated by a code generation model without explicitly executing the programs. Our rankers are fault-aware i.e., trained to predict the fine-grained classes of failure modes and we showed the effectiveness of the fault-aware rankers in improving pass@k and exec@k metrics for various code generation models and tasks.

One of the main limitations is that the CODERANKER approach is not sound, i.e., a ranker can classify a correct program as wrong and vice-versa. Additionally, we incur extra inference time to get better performance, because we now have to generate $n >> k$ programs to filter $k$ programs to show to the user. Our current approach also relies on sampling full programs from the code generation model before using CODERANKER. In the future, we want to investigate ranking/classifying partial programs, which can in-turn reduce the inference time by pruning out wrong programs early. Another future direction for our work is to investigate other ranker model architectures such as those that can better levarage the code structure and exploring other ranker tasks such as generating the full error message. Finally, it will be interesting to investigate the transferability of our CODERANKER approach to generic programming tasks (rather than just competition programming). A main challenge. here, is the lack of a generic programming dataset which needs to collected by scraping public sources such as GitHub. Our preliminary experiments on such a generic programming dataset show that existing code generation models usually produce more programs with execution errors rather than programs with intent errors . This observation combined with our results in this paper that show CODERANKER is usually better at identifying execution errors than intent errors, we are very optimistic that a fault aware CODERANKER approach would also improve the code generation for generic programming tasks.

**Acknowledgements:**    We thank Todd Mytkowicz, Piali Choudhury, Rahee Gosh Peshwaria, Curtis von Veh, and Xiaodong Liu for helpful discussions on this work.

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
