# A   Appendix

## A.1   More details on the fault-aware ranker datasets

Table 12 and Table 13 show additional examples for each execution error and intent error class.

Figure 6 shows the distribution of the various classes of execution and intent errors for the ranker dataset generated by the Codex model using the APPS training dataset.

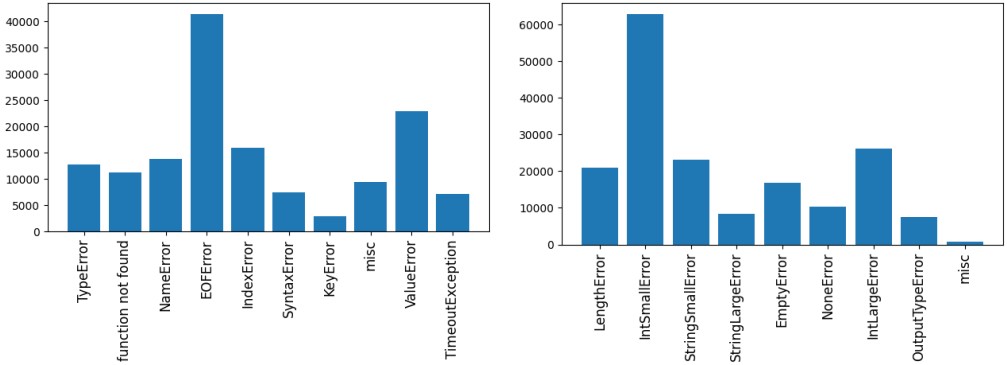

**Figure 6:** Distributions for various classes of execution errors (left) and intent errors (right) in the ranker training dataset obtained using the Codex code gen. model.

## A.2   Additional results

### A.2.1   Ablations on different ranker tasks

Table 14 and Table 15 show the various ranked metrics for rankers trained on different fault-aware tasks when using the Codex and GPT-Neo 125M code generation models respectively. Figure 7 shows 4 training curves, $X$, showcasing the validation ranked exec@1 curves for the model $X$ when combined with 5 different rankers $R^*_{D_X}$.

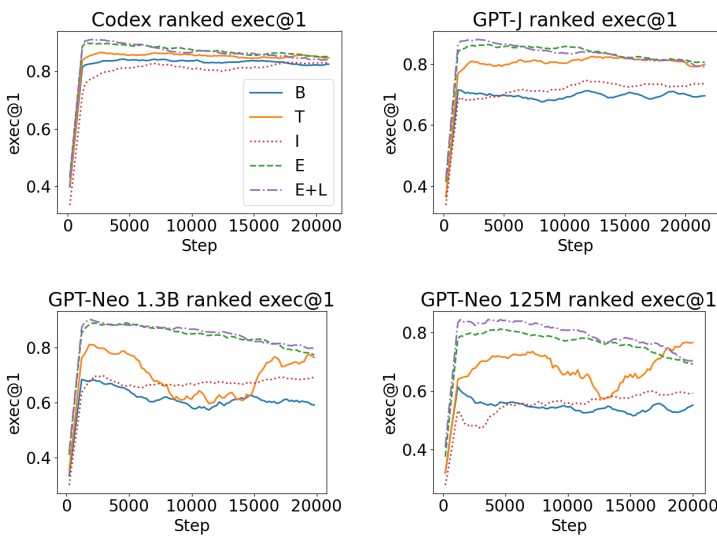

**Figure 7:** Training curves (smoothed) measuring ranked exec@1 on the validation set as training progresses for various rankers on different code generation models.

| Class | Example 1 | Example 2 |
|---|---|---|
| NameError | ```python
def bird_code(arr):
    if 'hyphen' in arr:
        return [arr[i][0] + ...
# name 'i' is not defined in line 2
``` | ```python
def args_to_string(args):
    return '\n'.join(sorted(s,
        key=lambda x: x.isdigit())) + '\n'
# name 's' is not defined in line 2
``` |
| ValueError | ```python
t=int(input())
for i in range(t):
    s,g,d,t=map(int,input().split())
    print(d-1)
    print(s*g)
# Too many values to unpack in
# line 2 (expected 4, got 5)
``` | ```python
def capitals_first(text):
    return '{:b}{}'.format(text.lower(),
        text.upper(), text.capitalize())
# Unknown format code 'b' for object
# of type 'str' in line 1
``` |
| EOFError | ```python
t = int(input())
for i in range(t):
    s = input()
    a = sum(map(int, input().split(' ')))
    print(a)
# EOF when reading a line at Line 3
# caused due to duplicated calls
# to input() within the for loop.
``` | ```python
_ = input()
s = input()
ans = 1
for i in range(1, len(s)):
    if s[i] == "?" and s[i - 1] == "?":
    ans = (ans * 2) % 1000003
# EOF when reading a line at Line 1
``` |
| TypeError | ```python
def solomons_quest(arr):
    return [abs(a) for a in arr]
# bad operand type for abs():
# 'list' at Line 1
``` | ```python
def negation_value(s, val):
    if val in s: return False
    return s[0]
# 'in <string>' requires string as
# left operand, not bool at Line 1
``` |
| IndexError | ```python
for _ in range(int(input())):
    n,m=map(int,input().split())
    l=[0]*n
    for i in range(n*m):
        l[i]+=1
# List index out of range at Line 4
``` | ```python
def get_last_digit(index):
    digits = [x for x in \
    range(1,index + 1) if x < 10]
    return digits[index]
# List index out of range at Line 3
``` |
| KeyError | ```python
def define_suit(card):
    a = {"3C": "clubs", "3D":
    "diamonds", "3H": "hearts", "3S":
    "spades"}
    return a[card]
# KeyError('QS') at Line 4
``` | ```python
def league_standings(teams):
    return {i+1: teams[-i-1] for
        i in range(len(teams))}
# KeyError(-1) at Line 1
``` |
| Function not found | ```python
def seemingly(string):
    ...
# The test module is expecting a
# function named "apparently"
``` | ```python
# Empty code
``` |
| TimeoutException | ```python
def pre_fizz(n):
    list = []
    while n >= 0:
        list.append(n)
        n = n//1
    return list
# runs into an infinite loop
``` | ```python
def fibonacci(n):
    return n if n in [0, 1] else
    fibonacci(n - 1) +
    fibonacci(n - 2)
# inefficient implementation
``` |
| SyntaxError | ```python
def game_winners(gryffindor,
slytherin):
    if slytherin == "yes":
        return "It's a draw!".
    ...
# Syntax error at Line 3 (extra .)
``` | ```python
def process_data(data):
    return data[0] * data[1] *
    data[2] for i in range(len(data))
# Syntax error at Line 1
# (list comprehension syntax)
``` |
| Misc | ```python
def duplicate_count(text):
    return sum(count for c, count in
    text.items() for count in {'a', 'A', 'B'})
# AttributeError at Line 2
# (str has no attribute items)
``` | ```python
for _ in range(int(input())):
    n = int(input())
    a = a+1
    ...
# UnboundLocalError at line 2
# local variable 'a' is referenced
# before assignment.
``` |

**Table 12:** Examples of different classes of execution errors

### A.2.2 Ablations on different ranker datasets

Table 16 shows the ablation results when training rankers on datasets collected from code generation models. These results show the improvements of rankers on the Codex code generation model. We can see that a ranker trained on completions from the Codex model is better than a ranker trained on completions from other models. We can also see that a ranker trained on the mixed-large dataset (which use all completions from all models) beats the ranker trained on Codex dataset by additional 2.7%.

| Class | Example 1 | Example 2 |
|---|---|---|
| NoneError | ```python
def consecutive_sum(num):
    sum = 0
    n = len(str(num))
    for i in range(n−1):
        if num % i == 0 and sum > 0:
            return sum
# return statement never gets triggered
``` | ```python
def start_smoking(bars, boxes):
    rem = boxes * 18 − bars * 10
    if rem != 1:
        print((rem + 4) // 5)
    else:
        print(0)
# no return statement
``` |
| EmptyError | ```python
def grabscrab(word, possible_words):
    if word in possible_words:
        return possible_words[word]
    else:
        return []
# Got [], expected ['first']
``` | ```python
def interleave(args):
    return [elem for elem in zip(args)
            if len(elem) < len(args)]
# Got [], expected
# [1, 'c', 2, 'd', 3, 'e']
``` |
| OutputTypeError | ```python
def diamonds_and_toads(sentence, fairy):
    if fairy == 'good':
        return sum(1 for i in sentence.split())
    elif fairy == 'evil':
        return sum(1 for i in sentence.split())
# Got 3, expected
# [{'ruby': 3, 'crystal': 2}])
``` | ```python
class Solution:
    def longestPrefix(self, s: str) −> str:
        s = list(s)
        s = [i for i,j in zip(s,s[::−1][0:])]
        return s[:−2]
# Got ['"', 'l', 'e', 'v', 'e'], expected '"'
``` |
| LengthError | ```python
def rotate(arr, n):
    return list(range(0,len(arr)+n,n))
# Got [0, 1, 2, 3], expected
# ['c', 'a', 'b']
``` | ```python
def diamonds_and_toads(sentence,fairy):
    return dict(zip('Ruby␣Crystal',
                (0, 2, 1, 2, 0)))
# Got {'R': 0, 'u': 2, 'b': 1, 'y': 2, ' ': 0},
# expected {'ruby': 3, 'crystal': 2}
``` |
| IntSmallError | ```python
def is_balanced(source, caps):
    return all(x.startswith(caps) or x == caps
               for x in source)
# Got False, expected True
``` | ```python
T = int(input())
for _ in range(T):
    N = int(input())
    print(bin(N).count('0'))
# Got [[2], [3]], expected [[1], [2]]
``` |
| IntLargeError | ```python
def missing_angle(h, a, o):
    return int((a*h+o)/2) if o==0
        else int(a*h+o)
# Got 300, expected 37
``` | ```python
class Solution:
    def findMedian(self, nums1, nums2):
        nums1.sort()
        nums2.sort()
        return float('−inf')
# Got −inf, expected 2.0
``` |
| StringSmallError | ```python
def smash(words):
    return ''.join(word for word in words)
# Got 'helloworld',
# expected ['hello world']
``` | ```python
def solve(st):
    a = st.find("a")
    return a if len(st) == 0
        else "".join(sorted(list(st)))[0]
# Got 'a', expected 'x'
``` |
| StringLargeError | ```python
def jumping_number(number):
    number = str(number)
    if len(number)!= 0:
        return "Not!!"
    else:
        return "Jumping!!"
# Got 'Not!!', expected 'Jumping!!'
``` | ```python
def spacify(string):
    #your code here
    return ''.join(sorted(string.split()))
# Got 'Pippi', expected 'P i p p i'
``` |

**Table 13:** Examples of different classes of intent errors

| Codex + ranker | ranked pass@1 | ranked pass@5 | ranked exec@1 |
|---|---|---|---|
| $R^B_{D_{Codex}}$ | 37.6 | 64.6 | 83.4 |
| $R^T_{D_{Codex}}$ | **39.6** | 63.5 | 87.0 |
| $R^I_{D_{Codex}}$ | 36.5 | **64.7** | 81.8 |
| $R^E_{D_{Codex}}$ | 38.6 | 64.0 | 87.5 |
| $R^{E+L}_{D_{Codex}}$ | 37.1 | 61.5 | **88.3** |

**Table 14:** Ablation of the different fault-aware ranker tasks using Codex as the code-generation model (on the APPS validation dataset).

### A.2.3 Alternate ranker architectures

We experimented with a GPT-Neo 125M decoder-based model for the ranker and found it to be performing suboptimal to a CodeBERT-based ranker (even though both models are roughly the same size). The GPT-Neo 125M ranker trained on the Codex ranker dataset ($R^T_{D_{codex}}$) achieved 36.8% ranked pass@1 when combined with the Codex model while the CodeBERT-based ranker achieved 39.6%.

| GPT-Neo 125M + ranker | ranked pass@1 | ranked pass@5 | ranked exec@1 |
|---|---|---|---|
| $R^{B}_{D_{\text{GPT-Neo 125M}}}$ | 4.2 | 7.7 | 56.0 |
| $R^{T}_{D_{\text{GPT-Neo 125M}}}$ | 5.7 | 10.4 | 78.4 |
| $R^{I}_{D_{\text{GPT-Neo 125M}}}$ | **6.5** | 11.4 | 58.9 |
| $R^{E}_{D_{\text{GPT-Neo125M}}}$ | 5.7 | 11.0 | 74.7 |
| $R^{E+L}_{D_{\text{GPT-Neo 125M}}}$ | **6.5** | **11.9** | **87.6** |

**Table 15:** Ablation of the different fault-aware ranker tasks using GPT-Neo 125M model as the code-generation model (on the APPS validation set).

| Codex + Ranker | ranked pass@1 | ranked exec@1 |
|---|---|---|
| $R^{\text{best}}_{D_{\text{Codex}}}$ | 39.6 | 87.0 |
| $R^{\text{best}}_{D_{\text{GPT-J}}}$ | 28.3 | 83.6 |
| $R^{\text{best}}_{D_{\text{GPT-Neo 1.3B}}}$ | 26.9 | 83.6 |
| $R^{\text{best}}_{D_{\text{GPT-Neo 125M}}}$ | 27.3 | 81.8 |
| $R^{\text{best}}_{D_{\text{mixed-small}}}$ | 39.1 | 88.1 |
| $R^{\text{best}}_{D_{\text{mixed-large}}}$ | **42.3** | **91.0** |

**Table 16:** Ablation of the different ranker datasets (using the best ranker task for each dataset) on Codex completions (on APPS validation set).

We further conducted an experiment with a GPT-Neo 1.3B model for the ranker. It took almost 4 days to complete 7 epochs of training with 16 V100 GPUs running in parallel. And still, it didn't reach the performance achieved by the CodeBERT model. GPT Neo 1.3B ranker model achieved ranked pass@1 of 38.6% while CodeBERT ranker achieved 39.6%. We stopped GPT Neo 1.3B ranker training at 7 epochs because the training became unstable at that point. On the other hand, a CodeBERT based ranker took only 12 hours to train for 30 epochs (with 16 GPUs).

Table 17 shows the ranked pass@1 and ranked exec@1 performance of these various ranker architectures and Figure 8 shows the smoothed training curves for these three rankers.

We believe that a ranker task is a code understanding task, hence starting with pre-trained encoders such as CodeBERT are most effective compared to starting with a pre-trained encode-decoder or decoder only code-generation models. We leave the full study about the impact of various ranker architectures to future work. We also do not anticipate that just bigger ranker models would fill in the gap between current ranked pass@1 and pass@100 since a ranker needs to implicitly learn to execute which is a hard task. For that, we might need new architectures for capturing code structure and execution. For this future exploration, we hope that our ranker tasks will serve as good benchmarks for evaluating such architectures.

| Codex + Ternary Ranker | ranked pass@1 | ranked exec@1 |
|---|---|---|
| CodeBert Ranker | **39.6** | 87.0 |
| GPT-Neo 125M Ranker | 36.8 | 84.4 |
| GPT-Neo 1.3B Ranker | 38.6 | **88.6** |

**Table 17:** Ablation of the different ranker architectures on Codex completions (on APPS validation set; trained on the ternary ranker dataset).

### A.2.4 Impact of example unit tests in the prompt

Following prior works, our experiments so far include a small number of unit tests in the task description of the problem (as seen in Figure 1) so that we can directly compare our results with prior works. Although our CODERANKER approach does not explicitly use these unit tests (since we do not assume that an execution environment is given), our rankers can use the given unit tests as part of their language/code comprehension ability to better solve the tasks. So, in this section, we conduct an ablation study to measure the impact of the presence of these unit tests in the prompt on the ranking ability of CODERANKER. We created a best-effort version of the APPS training/validation dataset

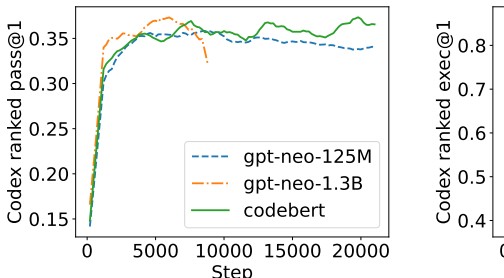

**Figure 8:** CodeBert vs GPT-Neo ranker models: Training curves showing ranked pass@1 and exec@1 on the APPS validation set with the ternary ranker.

that does not include the unit tests in the prompts and we trained a new ranker on this dataset. This new ranker only performs slightly worse than the ranker that is trained with the unit tests in the prompts, but it still significantly outperforms the approach without any ranker (see Table 18).

| Codex + Ternary Ranker | pass@1 | exec@1 |
|---|---|---|
| Prompts with example unit tests | **39.6** | 87.0 |
| Prompts without example unit tests | 38.0 | **89.1** |
| Without ranker | 26.0 | 69.7 |

**Table 18:** Ablation for evaluating the impact of example unit tests in the prompt.

### A.2.5 Comparison to ranking using the code gen. model's predicted probabilities

In this ablation, we compare the impact of ranking using a separate ranker model versus ranking using the code gen. model's predicted probabilities for the various generated programs. The results for the GPT-Neo 125M code gen. model can be found in Table 19; we can see that a separate CODERANKER produces the best ranking effect and in fact, using the intrinsic code gen. model's probabilities to rank results in a slightly worse performance compared to a no ranking approach.

| | pass@100 | pass@1 | pass@5 | exec@1 |
|---|---|---|---|---|
| GPT-Neo 125M | 23.6 | 1.4 | 5.2 | 41.1 |
| + ranking using generation probability | - | 1.0 | 3.7 | 43.6 |
| + CODERANKER | - | **6.5** | **11.4** | **58.9** |

**Table 19:** Comparing CODERANKER to ranking completions using Code Gen. model's predicted probabilities for the generated completions.

### A.2.6 Qualitative analysis of CODERANKER

In Table 20, we show various programs generated by the fine-tuned GPT-J model, the actual error message from the Python compiler upon executing these programs, and the various labels generated by the different CODERANKERs trained on various ranker tasks.

| Generated code | Actual error message | Predicted |
|---|---|---|
| ```python
def pillow(s):
    for i in range(len(s)):
        if s[i] == "n":
            if s[i+1:] == "B":
                return True
    return False
``` | Expected True, but got False | B: correct
T: intent error
I: IntSmallError
E: intent error
E+L: intent error at Line -1 |
| ```python
def camel_case(string):
    return "".join([x.capitalize() for
        x in string.split(" ")])
``` | Passed all tests | B: correct
T: intent error
I: StrSmallError
E: correct
E+L: correct at Line -1 |
| ```python
def sum_it_up(numbers_with_bases):
    return sum(int(str(n[0]) * n[1],
        base) for n in numbers_with_bases)
``` | NameError(base is not defined) at Line 1 | B: wrong
T: execution error
I: execution error
E: NameError
E+L: NameError at line 1 |
| ```python
for _ in range(int(input())):
    n = int(input())
    a = n - 2
    b = n - 1
    c = n
    if n % 2 == 0:
        print((a*((a-1)**2) + (b**2) -
        2*b + 2*c)%1000000007)
    else:
        print((a*((a-1)**2) + (c**2) -
        2*c + 2*b)%1000000007)
``` | Got [['7'], ['13']], but expected [['1'], ['3']] | B: wrong
T: intent error
I: IntLargeError
E: intent error
E+L: intent error at Line -1 |
| ```python
def bingo(array):
    s = "".join(array)
    return "WIN" if "BINGO" == s
        else "LOSE"
``` | TypeError(sequence item 0: expected str instance, int found) at Line 1 | B: correct
T: intent error
I: StringSmallError
E: intent error
E+L: TypeError at Line 1 |
| ```python
T = int(input())
for _ in range(T):
    n = int(input())
    S = sorted(list(map(int,
        input().split())))
    ans = 0
    for i in range(1, n+1):
        if S[i] > S[i-1]:
            ans += 1
            print(ans)
``` | Passed all tests | B: correct
T: execution error
I: execution error
E: IndexError
E+L: IndexError at Line 6 |

**Table 20:** We show various programs sampled from a fine-tuned GPT-J model, the actual error message, and the labels predicted by various ranker models $R_{\mathcal{D}_{GPTJ}}^{X}$. We use the green color to indicate that label is correct, red to indicate that the label is incorrect, and blue to indicate the case where the label correctly indicates an error in the program, but it indicates the wrong error type.