# OpenReview forum: "Fault-Aware Neural Code Rankers"
_NeurIPS.cc/2022/Conference — NeurIPS 2022 Accept_

### Official Review · Reviewer_DbiK · 2022-07-10

**Rating:** 7
**Confidence:** 4
**Soundness:** 4 excellent
**Presentation:** 4 excellent
**Contribution:** 2 fair

**Summary:**

This work trains a model to predict errors in generated code and rerank the generations. Current models are often evaluated using the pass@k metric, which assumes that the unit tests can be run on k generations to rank them. They motivate their method with these arguments:
1) Not test dependent (it could fail on unseen tests)
2) No burden to create tests
3) No need to install dependencies
4) No security risk linked to execution

Their model is trained on a dataset generated from APPS (where unit tests are available), but the model learns to find errors from the code only. They evaluate it for reranking the outputs of language models on APPS (validation and test datasets), HumanEval and MBPP.
They substantially improve the pass@1 scores of Codex, GPT-J, and GPT-NEO (1.3B and 125M params) on several benchmarks.


**Questions:**

* Did you do any ablation showing how much the dataset size impacts your performance in this context?
* Does the code generation task contain the example unit tests? If so, the ranker could use the unit tests to determine the error.



**Ethics Review Area:**

["I don’t know"]

**Limitations:**

The datasets they tested their method contain mostly functions from programming contests. It is fine, since it would be more difficult to test it on real data when no unit tests are available. However, they mention real usecases in the introduction (such as use in VSCode) and I believe that their method may not transfer as well to real data, with expected input/outputs that are more difficult to infer. I would have liked to see a discussion about that.

**Strengths And Weaknesses:**

Overall, I found the method in this paper simple, but their motivation is compelling, it is well-written, and it is well executed.
Strengths:
* Clear and well-written
* The motivation for this work is clear and compelling. Executing many generated snippets is often not doable in practice
* Even though their reranking is still far from perfect (as shown by the difference with the pass@5), they substantially improve the pass@1 for every model and dataset, showing the robustness of their method
* They give good insight into their model. The exec@1 improvement shows that execution errors are understandably easier to find.

Weaknesses:
* Limited technical novelty

---

> ### Author Response · Authors · 2022-08-02
> **Response**
>
> We thank the reviewer for their constructive comments. We are glad that the reviewer finds our paper simple, well-written, and has strong motivation.  We thank you for further comments on improving our paper and we answer your questions below:
>
> **Real world applications:**
>
> Our approach is very much applicable to generic programming (and not just limited to competition programming). Our preliminary experiments on a generic programming dataset scraped from GitHub show that existing code generation models usually produce more programs with execution errors rather than programs with intent errors  . This observation combined with our results in this paper that show rankers are usually better at identifying execution errors than intent errors, we are very optimistic that a fault aware ranker approach would also improve the code generation for generic programming tasks. In fact, our preliminary experiments show that a binary ranker trained on this GitHub dataset can improve pass@1 of Codex model from 42% to 47% (on a validation subset where pass@100 for Codex is 100%).
>
> Unfortunately, this generic programming dataset is a private dataset at this point, so we are unable to add results on that dataset yet.
>
> **Ablation for ranker dataset size:**
>
> For our mixed ranker dataset which combines the datasets from all code generation models, we did an ablation on two different dataset sizes (one is 4X the size of the other) Figure 3 and Table 15. Larger dataset produces slightly better results as expected (39.6% to 42.3% on D_codex). For detailed analysis on the impact of ranker dataset size, we would refer a reader to Cobbe et.al. 2021 , which did a great job at exploring the effect of dataset size on the rankers.
>
> Cobbe, Karl, et al. "Training verifiers to solve math word problems." arXiv preprint arXiv:2110.14168 (2021).
>
> **Unit tests in code generation task:**
>
> Some of our code generations do have a few examples in the prompt as shown in Figure 4 in the appendix. However, these unit tests are not expected, and they are not expected to be in any particular format. Our rankers do not explicitly use them since we don’t assume that an execution environment is provided (a sandbox with all dependencies installed), but our rankers can use the given examples as part of their language/code comprehension ability.

---

> > ### Comment · Reviewer_DbiK · 2022-08-08
> > **Response**
> >
> > Thank you for your answer.
> >
> > I understand that it would be difficult to test this method on real-world public data outside of the coding competition domain. Although they cannot be reproduced, the results of you preliminary experiment are interesting. Do you think you could mention them in the final version of the paper?
> >
> > I am satisfied with your answer on the ranker dataset size.
> >
> > About the unit tests, I agree that having the test cases in the text of the prompt is not the same as using an execution environment to test the output. I also believe that having a model that can use test cases described in a prompt can be useful. However, part of the motivation for your method is that it doesn't require the user to create tests. I believe it would be interesting to know how your method performs without test cases in the prompt.

---

> > > ### Author Response · Authors · 2022-08-08
> > > **Thank you**
> > >
> > > Thank you for the follow-up.
> > >
> > > We are glad to hear that you find the preliminary results on the real world dataset interesting. We will add a discussion about the real-world dataset to the paper. We will aim to talk about the challenges (such as the dataset collection) and expected insights (such as observing more execution errors than intent errors), but we think leaving the results for a follow up paper will be better for the community so that we can do thorough study and also try to make the results reproducible.
> > >
> > > Regarding unit tests, we agree that it is useful to look at the impact of the test cases in the prompt for both code generation and code ranking. We will setup an experiment for this and add a case study in the final version. One of the main reasons we included the test cases in the prompt in our experiments is to keep the prompts same as in prior works so that the results are comparable.

---

### Official Review · Reviewer_SQmW · 2022-07-11

**Rating:** 6
**Confidence:** 4
**Soundness:** 3 good
**Presentation:** 4 excellent
**Contribution:** 3 good

**Summary:**

This paper complements a deep learning based program synthesis module with a separate fault aware ranker framework that learns execution pattern of the generated program without actually executing it. The work is inline with recent advancements in large language models that show the inherent understanding of the problem by the model but its inability to search for the correct user intent based on the first greedily sampled decoder output. Hence it is necessary to use complex search strategy, which in the case of this paper happen to be another neural network. The results shown in the paper show consistent improvements in potential user experience.

**Questions:**

1) It will be great to understand the effectiveness of the ranker framework in detecting user bugs in held-out datasets. This will validate the robustness of the ranker framework. Are there any preliminary experiments in that direction?

2) How will the trained model on CodeBERT compare against using some baseline methods, some generative models were mentioned in the related literature. Can they be compared? If not, is it possible to fine-tune any autoregressive model on this task and compare against the current architecture?



**Limitations:**

The authors have adequately addressed the limitations. Some preliminary analysis on the limitations discussed can benefit the reader in understanding the robustness and current state of the error detection module developed by the authors.

**Strengths And Weaknesses:**

The paper has some noteworthy strengths. They are the first to effectively use neural techniques to classify intent errors for general purpose programs and do not restrict themselves to simpler domains. This is an important finding as it can lead the community into building better models for detecting code errors, which can be used to judge model generated code as well as human written code. They set a strong ground work for detecting code errors using deep learning models and show the potential of such approach by experimenting over a diverse set of domains and diverse set of SOTA models. Finally they show the effectiveness of complementing such synthesis models with ranker systems, thereby improving their applicability in real world scenarios.

The paper show the effectiveness of the approach in detecting bugs in model generated code thereby improving those systems. The presented results are to some extent expected, given that the model is augmenting the strong generative models with additional data of detecting intent errors. The system is also developed with the motivation of detecting errors early when they are generated by a generative model such to avoid unnecessary scenarios resulting from executing buggy codes. It is not exactly clear if running the system in a sandbox environment will completely remove any necessity of using this approach. Finally, the experiments were run solely with the purpose of improving the performance of code generation models, it will be interesting to find the effectiveness of the ranker in detecting bugs on unseen real code.

---

> ### Author Response · Authors · 2022-08-02
> **Response**
>
> We thank the reviewer for their valuable comments. We are very encouraged that you appreciate our motivation, technical contribution, and evaluation. We also thank you for bringing up several interesting ways to further improve the paper.  Below, we answer the following questions:
>
> **Detecting bugs on real code:**
>
> This is a great suggestion, and it is indeed one of our future directions. It will be interesting to see if our ranker dataset can complement a real bugs dataset (which might be very hard to collect).
>
> That said, a key requirement for fault detectors on real code is to reduce the false positive rate (we do not want to tell a user that their code is wrong when it is not). Many of the current approaches (including ours) have a significantly high false positive rate.  However, a high false positive rate is not an issue for a ranker as long as the ranker is able to rank at least one correct code high.  So, we would expect the motivations and approaches for the two applications will be slightly different.
>
> **Other ranker model architectures:**
>
> We experimented with a GPT-Neo 125M model for the ranker and found it to be performing suboptimal to a CodeBERT-based ranker (even though both models are roughly the same size). The GPT-Neo 125M model achieved 36.8% ranked pass@1 on the Codex ranker dataset ($R^T_{D_{codex}}$) while CodeBERT achieved 39.6% (Table 4 in the paper). Because the ranker task is a code understanding task (as opposed to a code generation task), pre-trained encoder models such as CodeBERT are both more efficient and more effective compared to pre-trained encode-decoder or decoder only models that are designed for code generation tasks. We added this comparison to the new appendix (Figure 7) and show a training curve plot for the two models.
>
> We did not consider bigger models such GPT-Neo 1.3B model because finetuning them is almost 20 -30X slower than finetuning a CodeBERT model. We chose CodeBERT for this paper because it allowed us to quickly experiment how the different fine-grained ranker tasks would impact the ranking performance (which is the main goal for this paper).
>
> As our paper focuses on studying how to harness code rankers to improve code generation performance, we leave further  study about the impact of different ranker architectures to future work. We also do not anticipate that simply increasing ranker model size would fill in the gap between current ranked pass@1 and pass@100, since it implicitly requires the ranker to learn to execute (which is a much harder task). As a result, we might need new architectures for capturing code structure and execution. For this future exploration, we believe that our ranker tasks will serve as good benchmarks for evaluating such architectures.
>
> **Executing in a sandbox vs neural rankers:**
>
> We don’t expect that just executing in a sandbox will absolve the necessity of a neural ranker in practical scenarios because most development code is incomplete  (does not have all the dependencies), not executable, and lack unit tests. In addition, executing code at inference time is very time consuming compared to inferencing a neural ranker (our neural ranker reduces this overhead).

---

> > ### Comment · Reviewer_SQmW · 2022-08-09
> > **Thank you**
> >
> > Thank you for addressing the main concerns. Looking forward to future extensions of the work.

---

### Official Review · Reviewer_xLPt · 2022-07-21

**Rating:** 7
**Confidence:** 4
**Soundness:** 3 good
**Presentation:** 3 good
**Contribution:** 3 good

**Summary:**

This paper proposes to train fault-aware neural code rankers (I would prefer to call them classifiers) that can predict the correctness of sample programs without executing them (using unit tests). The rankers are trained to predict different execution information, such as compile/runtime error types (e.g., an IndexError or a TypeError). The paper shows that the trained rankers can significantly increase the pass@k accuracy of various code generation models, e.g., Codex, GPT-Neo, GPT-J on popular benchmarks, e.g., HumanEval, APPS, and MBPP.

**Questions:**

1. Given that the trained rankers are actually classifiers, how exactly the top-1 candidate is chosen to compute ranked pass@1? I assume the ranking is done based on code generation models provided likelihood score, and then wrong samples were filtered out based on the predictions of the fault-aware classifiers. I think this is an important piece of information that was not clarified.

2. Out of curiosity asking this question, since the experiments were conducted on V100-32GB GPUs, what confined the authors to choose better models than CodeBERT (e.g., CodeT5, PLBART) for the ranking models? In fact, the authors could have used GPT-Neo 1.3B as the rankers.

**Limitations:**

The major limitation of this work is the conflict between the motivation of the work and what is being proposed. While the paper motivates that in the real world, we cannot assume that there will be unit tests readily available to evaluate models via execution. However, the paper used the APPS dataset composed of unit tests. The authors didn't discuss how their work could be leveraged in the real world. For example, can we use the authors' proposed idea for generic programming (not competitive programming only)? I would prefer to see some discussion.

**Strengths And Weaknesses:**

**Significance**
The paper pointed out that current literature assumes that the unit tests are generally available to evaluate code generation models via execution and it is safe to execute the generated programs. The paper argues that such assumptions are impractical in real-world software development, making the work significant to investigate alternatives.

**Originality**
While I am unaware of any prior works that exactly do the same (training rankers to predict errors), the proposed idea is intuitive (anyone can come up with the idea with careful thinking). Though I see this work's technical contribution as thin; I appreciate the author's effort to study the effectiveness of the proposed approach.

**Clarity**
The paper is written fairly clearly. Except for one crucial piece of information, I did not feel anything important was missed. Also, the paper should rather use classifiers in place of rankers.

**Quality**
Overall the quality of the work is good. However, I felt the authors didn't go deeper in investigating stronger code representation models (they chose CodeBERT, which is a weaker model in literature; I would love to see a  few more models being investigated as rankers). This was important since there is a significant difference between pass@100 and ranked pass@1 on the benchmarks.

---

> ### Author Response · Authors · 2022-08-02
> **Response**
>
> We thank the reviewer for their thoughtful review. We are glad that you find our work well motivated and intuitive. We also appreciate the constructive suggestions that we clarify below.
>
> **Other ranker model architectures:**
>
> We experimented with a GPT-Neo 125M model for the ranker and found it to be performing suboptimal to a CodeBERT-based ranker (even though both models are roughly the same size). The GPT-Neo 125M model achieved 36.8% ranked pass@1 on the Codex ranker dataset ($R^T_{D_{codex}}$) while CodeBERT achieved 39.6% (Table 4 in the paper). Because the ranker task is a code understanding task (as opposed to a code generation task), pre-trained encoder models such as CodeBERT are both more efficient and more effective compared to pre-trained encode-decoder or decoder only models that are designed for code generation tasks. We added this comparison to the new appendix (Figure 7) and show a training curve plot for the two models.
>
> We did not consider bigger models such GPT-Neo 1.3B model because finetuning them is almost 20 -30X slower than finetuning a CodeBERT model. We chose CodeBERT for this paper because it allowed us to quickly experiment how the different fine-grained ranker tasks would impact the ranking performance (which is the main goal for this paper).
>
> As our paper focuses on studying how to harness code rankers to improve code generation performance, we leave further  study about the impact of different ranker architectures to future work. We also do not anticipate that simply increasing ranker model size would fill in the gap between current ranked pass@1 and pass@100, since it implicitly requires the ranker to learn to execute (which is a much harder task). As a result, we might need new architectures for capturing code structure and execution. For this future exploration, we believe that our ranker tasks will serve as good benchmarks for evaluating such architectures.
>
> **Ranked pass@1 computation:**
>
> We use the probability of predicting a code sample as “CORRECT”  by the ranker model as the score for sorting the code samples. This probability is extracted using the real values from the last layer before the SoftMax layer. We do not use the probabilities computed by the code generation model as they are not very effective for ranking. We explain this in the beginning of section 3, but we will aim to clarify it further.
>
> **Real world applications:**
>
> Our approach is very much applicable to generic programming (and not just limited to competition programming). Our preliminary experiments on a generic programming dataset scraped from GitHub show that existing code generation models usually produce more programs with execution errors rather than programs with intent errors  . This observation combined with our results in this paper that show rankers are usually better at identifying execution errors than intent errors, we are very optimistic that a fault aware ranker approach would also improve the code generation for generic programming tasks. In fact, our preliminary experiments show that a binary ranker trained on this GitHub dataset can improve pass@1 of Codex model from 42% to 47% (on a validation subset where pass@100 for Codex is 100%).
>
> Unfortunately, this generic programming dataset is a private dataset at this point, so we are unable to add results on that dataset yet.

---

> > ### Comment · Reviewer_xLPt · 2022-08-09
> > **Thank you for the responses**
> >
> > I cannot accept the argument of not exploring larger models for the sake of training time, even though the experiments were performed in V100 GPUs. What was the training time for CodeBERT? CodeBERT is a 125M parameter model while GPT-Neo is 1.3B. How many V100-32GB GPUs were involved in this work? The paper mentions that GPT-J and GPT-Neo code generation models were trained on the APPS dataset for 2 epochs with a batch size of 256. Then why GPT-Neo 1.3B cannot be trained as a re-ranker?
> >
> > I am emphasizing the model size of the re-ranker because currently, there is a big gap between ranked pass@k (k=1, 5) and pass@100, and I firmly believe this gap could be reduced by increasing the model size. I am not saying we do not need further study on ranker structure OR increasing model size is the ultimate solution. However, I think setting up a strong baseline is particularly important because soon, I do not want to see another work without significant technical differences but just with a large re-ranker outperforming this work.
> >
> > I am increasing my score by 1 point as the authors address my two other questions.

---

> > > ### Author Response · Authors · 2022-08-09
> > > **Response**
> > >
> > > Your concerns about setting a strong baseline is valid and we did set up a run with GPT-Neo 1.3B ranker before our rebuttal response for our own curiosity. It took almost 4 **days** to complete 7 epochs of training with 16 V100 GPUs running in parallel. And still, it didn't reach the performance achieved by the CodeBERT model. GPT Neo 1.3B model achieved ranked pass@1 of 38.6% while CodeBERT achieved 39.6%. We will add these results to Figure 7 in the appendix. We stopped GPT Neo 1.3B ranker training at 7 epochs because the training became unstable at that point.
> > >
> > > On the other hand, a CodeBERT based ranker took only 12 **hours** to train for 30 epochs (with 16 GPUs).
> > >
> > > The difference between training for code generation task and training for ranker task is that the ranker dataset is much bigger in size almost (50X) because we collect 100 completions for each code generation task and the ranker training takes significantly more epochs (15X) to converge.

---

> > > > ### Comment · Reviewer_xLPt · 2022-08-09
> > > > **Thank you for the response.**
> > > >
> > > > I appreciate these details. These are indeed important. I am increasing my score. Thank you.

---

> > > > > ### Author Response · Authors · 2022-08-09
> > > > > **Thank you**
> > > > >
> > > > > Thank you, we are very glad that your suggestions helped us to perform a useful experiment!

---

### Meta-Review · Area_Chair_rz3j · 2022-08-25

**Recommendation:** Accept
**Confidence:** Certain

**Metareview:**

The paper was well-received. The main idea is fairly simple, but the problem is important and the writing and empirical evaluation are solid. Based on the reviewers' advice, I am recommending acceptance. Please make sure to incorporate the reviewer feedback into the final version.


**Award:**

No

---

### Decision · Program_Chairs · 2022-09-14

Accept